# Universal Domain Adaptation through Self-Supervision

**Kuniaki Saito**[1]        **Donghyun Kim**[1]        **Stan Sclaroff**[1]

**Kate Saenko**[1,2]
[1]Boston University [2]MIT-IBM Watson AI Lab

[keisaito,dohnk,sclaroff,saenko]@bu.edu

## Abstract

Unsupervised domain adaptation methods traditionally assume that all source categories are present in the target domain. In practice, little may be known about the category overlap between the two domains. While some methods address target settings with either partial or open-set categories, they assume that the particular setting is known a priori. We propose a more universally applicable domain adaptation framework that can handle arbitrary category shift, called Domain Adaptive Neighborhood Clustering via Entropy optimization (DANCE). DANCE combines two novel ideas: First, as we cannot fully rely on source categories to learn features discriminative for the target, we propose a novel neighborhood clustering technique to learn the structure of the target domain in a self-supervised way. Second, we use entropy-based feature alignment and rejection to align target features with the source or reject them as unknown categories based on their entropy. We show through extensive experiments that DANCE outperforms baselines across open-set, open-partial, and partial domain adaptation settings. Implementation is available at https://github.com/VisionLearningGroup/DANCE.

## 1  Introduction

Deep neural networks can learn highly discriminative representations for image recognition tasks [8, 37, 20, 31, 16], but do not generalize well to new domains that are not distributed identically to the training data. Domain adaptation (DA) aims to transfer representations of source categories to novel target domains without additional supervision. Recent deep DA methods primarily achieve this by minimizing the feature distribution shift between the source and target samples [11, 24, 38]. However, these methods make strong assumptions about the degree to which the source categories overlap with the target domain, which limits their applicability to many real-world settings.

In this paper, we investigate the problem of *Universal DA*. Suppose $L_s$ and $L_t$ are the label sets in the source and target domain. In Universal DA we want to handle all of the following potential "category shifts": closed-set ($L_s = L_t$), open-set ($L_s \subset L_t$) [1, 35], partial ($L_t \subset L_s$) [2], or a mix of open and partial [43]. Existing DA methods cannot address Universal DA well because they are each designed to handle just one of the above settings. However, since the target domain is unlabeled, we may not know in advance which of these situations will occur. Thus, an unexpected category shift could lead to catastrophic misalignment. For example, using a closed-set method when the target has novel ("unknown") classes could incorrectly align them to source ("known") classes. The underlying issue at play is that existing work heavily relies on prior knowledge about the category shift.

The second problem is that the over-reliance on source supervision makes it challenging to obtain discriminative features on the target. Prior methods focus on aligning target features with source, rather than on exploiting structure specific to the target domain. In the universal DA setting, this

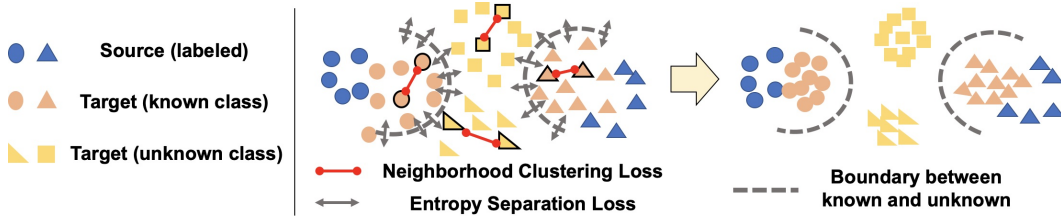

Figure 1: We propose *DANCE*, which combines a self-supervised clustering loss (red) to cluster neighboring target examples and an entropy separation loss (gray) to consider alignment with source (best viewed in color).

means that we may fail to learn features useful for discriminating "unknown" categories from the known categories, because such features may not exist in the source. Self-supervision was proposed in [5] to extract domain-generalizable features, but it is limited in that they did not utilize the cluster structure of the target domain.

We propose to overcome these challenging problems by introducing *Domain Adaptive Neighborhood Clustering via Entropy optimization (DANCE)*. An overview is shown in Fig. 1. Rather than relying only on the supervision of source categories to learn a discriminative representation, *DANCE* harnesses the cluster structure of the target domain using self-supervision. This is done with a "neighborhood clustering" technique that self-supervises feature learning in the target. At the same time, useful source features and class boundaries are preserved and adapted via distribution alignment with batch-normalization [7] and a novel partial domain alignment loss that we refer to as "entropy separation loss". This loss allows the model to either match each target example with the source, or reject it as an "unknown" category. Our contributions are summarized as follows: (i) we propose *DANCE*, a universal domain adaptation framework that can be applied out-of-the-box without prior knowledge of specific category shift, (ii) design two novel loss functions, neighborhood clustering and entropy separation, for category shift-agnostic adaptation, (iii) experimentally observe that *DANCE* is the only method that outperforms the source-only model in every setting, achieving state-of-the-art performance on all open-set and open-partial DA settings, and some partial DA settings, and (iv) learn discriminative features of "unknown" target samples without any supervision.

## 2   Related Work

**Closed-set Domain Adaptation (CDA).** The main challenge in domain adaptation (DA) is to leverage unlabeled target data to improve the source classifier's performance while accounting for domain shift. Classic approaches measure the distance between feature distributions in source and target, then train a model to minimize this distance. Many DA methods utilize a domain classifier to measure the distance [11, 39, 24, 25], while others minimize utilize pseudo-labels assigned to target examples [34, 47]. Clustering-based methods are proposed by [9, 36, 15]. These and other mainstream methods assume that all target examples belong to source classes. In this sense, they rely heavily on the relationship between source and target. **Partial Domain Adaptation (PDA)** handles the case where the target classes are a subset of source classes. This task is solved by performing importance-weighting on source examples that are similar to samples in the target [2, 44, 3]. **Open set Domain Adaptation (ODA)** deals with target examples whose class is different from any of the source classes [28, 35, 23]. The drawback of ODA methods is that they assume we necessarily have unknown examples in the target domain, and can fail in closed or partial domain adaptation. The idea of **Universal Domain Adaptation (UniDA)** was proposed in [43]. However, they applied their method to a mixture of PDA and ODA, which we call **OPDA**, where the target domain contains a subset of the source classes plus some unknown classes. Our goal is to propose a method that works well on CDA, ODA, PDA, and OPDA. We call the task UniDA in our paper. **Entropy minimization** [13] for unlabeled samples is popular in semi-supervised learning. In CDA, its effectiveness is confirmed when combined with batch-normalization based domain alignment [4]. Pseudo-labeling is a similar approach since it increases the confidence of the prediction for target samples and also performs well on CDA when combined with domain-specific batch-normalization [7]. These methods simply attempt to increase the confidence for "known" classes while our entropy separation loss can decrease the confidence to reject "unknown" classes.

**Self-Supervised Learning.** Self-supervised learning obtains features useful for various image recognition tasks by using a large number of unlabeled images [10]. A model is trained to solve a pretext (surrogate) task such as solving a jigsaw puzzle [27] or instance discrimination [41].

[6] trained a model to predict each sample's cluster index given by k-means clustering. Directly applying the method to UniDA is challenging, since we need to know the number of clusters in the target domain. [19, 46] proposed to perform instance discrimination and trained a model to discover neighborhoods for each example. Their result indicates that we can cluster samples without specifying their cluster centers. They calculate cross entropy loss on the probabilistic distribution of similarity between examples. Our neighborhood clustering (NC) is similar in that we aim to perform unsupervised clustering of unlabeled examples without specifying cluster centers, but different in that [19, 46] require assigning neighbors for each example. Instead, we perform entropy minimization on the similarity distribution among unlabeled target examples and source prototypes. Objectives that can cluster samples without specifying the center can possibly replace NC. But, note that one of our contributions is to provide a framework for UniDA that exploits cluster structure of the target domain and domain-alignment objective with a rejection option. Since these methods are not designed for DA, we focus on the comparison with DA baselines in our main paper. We provide the ablation study of replacing our neighborhood clustering loss with [19, 6, 5] to better understand each loss in supplemental material.

## 3   DANCE: Domain Adaptive Neighborhood Clustering via Entropy optimization

Our task is universal domain adaptation: given a labeled source domain $\mathcal{D}_s = \{(\mathbf{x}_i^s, y_i{}^s)\}_{i=1}^{N_s}$ with "known" categories $L_s$ and an unlabeled target domain $\mathcal{D}_t = \{(\mathbf{x}_i^t)\}_{i=1}^{N_t}$ which contains all or some "known" categories and possible "unknown" categories. Our goal is to label the target samples with either one of the $L_s$ labels or the "unknown" label. We train the model on $\mathcal{D}_s \cup \mathcal{D}_t$ and evaluate on $\mathcal{D}_t$. We seek a truly universal method that can handle any possible category shift without prior knowledge of it. The key is not to force complete alignment between the entire source and target distributions, as this may result in catastrophic misalignment. Instead, the challenge is to extract well-clustered target features while performing a relaxed alignment to the source classes and potentially rejecting "unknown" points.

We adopt a prototype-based classifier that maps samples close to their true class centroid (prototype) and far from samples of other classes. We first propose to use self-supervision in the target domain to cluster target samples. We call this technique **neighborhood clustering (NC)**. Each target point is aligned either to a "known" class prototype in the source or to its neighbor in the target. This allows the model to learn a discriminative metric that maps a point to its semantically close match, whether or not its class is "known". This is achieved by minimizing the entropy of the distribution over point similarity. Second, we propose an **entropy separation loss (ES)** to either align the target point with a source prototype or reject it as "unknown". The loss is applied to the entropy of the "known" category classifier's output to force it to be either low (the sample should belong to a "known" class) or high (the sample should be far from any "known" class). In addition, we utilize domain-specific batch normalization [7, 22, 33] to eliminate domain style information as a form of weak domain alignment.

### 3.1   Network Architecture

We adopt the architecture used in [33], which has an L2 normalization layer before the last linear layer. We can regard the weight vectors in the last linear layer as prototype features of each class. This architecture is well-suited to our purpose of finding a clustering over both target features and source prototypes. Let $G$ be the feature extraction network which takes an input $\mathbf{x}$ and outputs a feature vector $\boldsymbol{f}$. Let $\mathbf{W}$ be the classification network which consists of one linear layer without bias. The layer consists of weight vectors $[\mathbf{w}_1, \mathbf{w}_2, \ldots, \mathbf{w}_K]$ where $K$ represents the number of classes in the source. $\mathbf{W}$ takes L2 normalized features and outputs $K$ logits. $\boldsymbol{p}$ denotes the output of $\mathbf{W}$ after the softmax function.

### 3.2   Neighborhood Clustering (NC)

The principle behind our self-supervised clustering objective is to move each target point either to a "known" class prototype in the source or to its neighbor in the target. By making nearby points closer, the model learns well-clustered features. If "unknown" samples have similar characteristics with other "unknown" samples, then this clustering objective will help us extract discriminative features. This intuition is illustrated in Fig. 1. The important point is that we do not rely on strict distribution alignment with the source in order to extract discriminative target features. Instead we propose to minimize the entropy of each target point's similarity distribution to other target samples and to prototypes. To minimize the entropy, the point will move closer to a nearby point (we assume

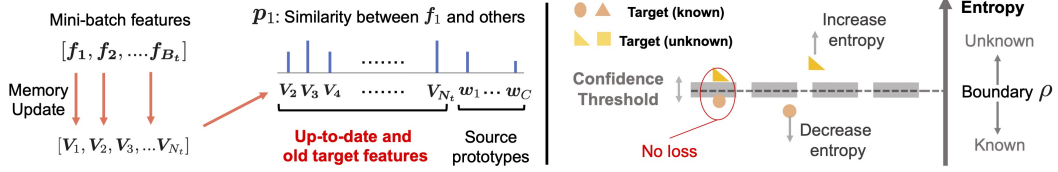

Figure 2: **Left**: Similarity distribution calculation in neighborhood clustering (best viewed in color). We minimize the entropy of the similarity distribution between each point (as shown for $f_1$), the prototypes, and the other target samples. Since most target samples are absent in the mini-batch, we store the their features in a memory bank, updating it with each batch. **Right**: An overview of the entropy separation loss (best viewed in color). We further decrease small entropy to move the sample to a "known"-class prototype, and increase large entropy to move it farther away. Since distinguishing "known" vs "unknown" samples near the boundary is hard, we introduce a confidence threshold that ignores such ambiguous samples.

a neighbor exists) or to a prototype. The advantage of the design is that we do not have to specify the number of clusters in the target domain, which is suitable for Universal DA. This approach is illustrated in Fig. 2.

Specifically, we calculate the similarity to all target samples and prototypes for each mini-batch of target features. Let $V \in R^{N_t \times d}$ denotes a memory bank which stores all target features and $F \in R^{(N_t+K) \times d}$ denotes the target features in the memory bank and the prototype weight vectors, where $d$ is the feature dimension in the last linear layer:

$$V = [V_1, V_2, ...V_{N_t}], \tag{1}$$

$$F = [V_1, V_2, ...V_{N_t}, \mathbf{w}_1, \mathbf{w}_2, \ldots, \mathbf{w}_K], \tag{2}$$

where the $V_i$ and $\mathbf{w}$ are L2-normalized. To consider target samples absent in the mini-batch, we employ a memory bank to store and use the features to calculate the similarity as done in [41]. In every iteration, $V$ is updated with the mini-batch features. Let $f_i$ denote features in the mini-batch and $B_t$ denote sets of target samples' indices in the mini-batch. For all $i \in B_t$, we set

$$V_i = f_i. \tag{3}$$

Therefore, the memory bank $V$ contains both updated target features from the current mini-batch and the older target features absent in the mini-batch. Unlike [41], we update the memory so that it simply stores features, without considering the momentum of features in previous epochs. Let $F_j$ denote the $j$-th item in $F$, then the probability that the feature $f_i$ is a neighbor of the feature or prototype $F_j$ is, for $i \neq j$ and $i \in B_t$,

$$p_{i,j} = \frac{\exp(F_j^\top f_i / \tau)}{Z_i}, \tag{4}$$

where

$$Z_i = \sum_{j=1, j \neq i}^{N_t+K} \exp(F_j^\top f_i / \tau), \tag{5}$$

and the temperature parameter $\tau$ controls the distribution concentration degree [18]. Therefore, $\tau$ controls the number of neighbors for each sample, which implicitly affects the number of clusters. We provide the analysis on the parameter in supplemental material. Then, the entropy is calculated as

$$\mathcal{L}_{\text{nc}} = -\frac{1}{|B_t|} \sum_{i \in B_t} \sum_{j=1, j \neq i}^{N_t+K} p_{i,j} \log(p_{i,j}). \tag{6}$$

We minimize the above loss to align each target sample to either a target neighbor or a prototype, whichever is closer.

### 3.3 Entropy Separation loss (ES)

The neighborhood clustering loss encourages the target samples to become well-clustered, but we still need to align some of them with "known" source categories while keeping the "unknown" target samples far from the source. In addition to the domain-specific batch normalization (see Sec. 3.4), which can work as a form of weak domain alignment, we need an explicit objective to encourage

Table 1: Summary of the Universal comparisons. Each dataset (Office, OC, OH, VisDA) has multiple domains and adaptation scenarios and we provide the average accuracy over all scenarios. Our DANCE method substantially improves performance compared to the source-only model in all settings and the average rank of DANCE is significantly higher than all other baselines.

| Method | Closed DA | | | Partial DA | | | Open set DA | | | Open-Partial DA | | | Avg | |
| | Office | OH | VD | OC | OH | VD | Office | OH | VD | Office | OH | VD | Acc | Rank |
|---|---|---|---|---|---|---|---|---|---|---|---|---|---|---|
| Source Only | 76.5 | 54.6 | 46.3 | 75.9 | 57.0 | 49.9 | 89.1 | 69.6 | 43.2 | 86.4 | 71.0 | 38.8 | 61.7 | 4.8± 1.2 |
| DANN [12] | **85.9** | 62.7 | 69.1 | 42.2 | 40.9 | 38.7 | 88.7 | 72.8 | 48.2 | 88.7 | 76.7 | 50.6 | 65.7 | 3.5± 1.7 |
| ETN [3] | 85.2 | 64.0 | 64.1 | **92.8** | 69.4 | 59.8 | 88.2 | 71.9 | 51.7 | 88.3 | 72.6 | 66.6 | 70.5 | 2.9± 1.6 |
| STA [23] | 73.6 | 44.7 | 48.1 | 69.8 | 47.9 | 48.2 | 89.9 | 69.3 | 48.8 | 89.8 | 72.6 | 47.4 | 61.2 | 4.5± 1.3 |
| UAN [43] | 84.4 | 58.8 | 66.4 | 52.9 | 34.2 | 39.7 | 91.0 | 74.6 | 50.0 | 84.1 | 75.0 | 47.3 | 62.0 | 4.1± 1.3 |
| DANCE (ours) | 85.5 | **69.1** | **70.2** | 84.7 | **71.1** | **73.7** | **94.1** | **78.1** | **65.3** | **93.9** | **80.4** | **69.2** | **77.3** | **1.2± 0.4** |

alignment or rejection of target samples. As pointed out in [43], "unknown" target samples are likely to have a larger entropy of the source classifier's output than "known" target samples. This is because "unknown" target samples do not share common features with "known" source classes.

Inspired by this, we propose to draw a boundary between "known" and "unknown" points using the entropy of a classifier's output. We visually introduce the idea in Fig. 2. The distance between the entropy and threshold boundary, $\rho$, is defined as $|H(\boldsymbol{p}) - \rho|$, where $\boldsymbol{p}$ is the classification output for a target sample. By maximizing the distance, we can make $H(\boldsymbol{p})$ far from $\rho$. We expect that the entropy of "unknown" target samples will be larger than $\rho$ whereas for the "known" ones it will be smaller. Tuning the parameter $\rho$ based on each adaptation setting requires a validation set. Instead, we define $\rho = \frac{\log(K)}{2}$, where $K$ is the number of source classes. Since $\log(K)$ is the maximum value of $H(\boldsymbol{p})$, we assume $\rho$ depends on it, and confirm that the defined value empirically works well. We perform an analysis of $\rho$ in the supplemental material. The above formulation assumes that "known" and "unknown" target samples can be separated with $\rho$. However, in many cases, the threshold can be ambiguous and can change due to domain shift. Therefore, we propose to introduce a confidence threshold parameter $m$ such that the final form of the loss is

$$\mathcal{L}_{es} = \frac{1}{|B_t|} \sum_{i \in B_t} \mathcal{L}_{es}(\boldsymbol{p}_i), \quad \mathcal{L}_{es}(\boldsymbol{p}_i) = \begin{cases} -|H(\boldsymbol{p}_i) - \rho| & (|H(\boldsymbol{p}_i) - \rho| > m), \\ 0 & otherwise. \end{cases} \quad (7)$$

The introduction of the confidence threshold $m$ allows us to give the separation loss only to confident samples. When $|H(\boldsymbol{p}_i) - \rho|$ is sufficiently large, the network is confident about a decision of "known" or "unknown". Thus, we train the network to make the sample far from the value $\rho$.

### 3.4 Training with Domain Specific Batch Normalization

To enhance alignment between source and target domain, we propose to utilize domain-specific batch normalization [7, 22, 33]. The batch normalization layer whitens the feature activations, which contributes to a performance gain. As reported in [33], simply splitting source and target samples into different mini-batches and forwarding them separately helps alignment. This kind of weak alignment matches our goal because strongly aligning feature distributions can harm the performance on non-closed set domain adaptation.

**Final Objective**. The final objective is

$$\mathcal{L} = \mathcal{L}_{cls} + \lambda(\mathcal{L}_{nc} + \mathcal{L}_{es}), \quad (8)$$

where $\mathcal{L}_{cls}$ denotes the cross-entropy loss on source samples. The loss on source and target is calculated in a different mini-batch to achieve domain-specific batch normalization. To reduce the number of hyper-parameters, we used the same weighting hyper-parameter $\lambda$ for $\mathcal{L}_{nc}$ and $\mathcal{L}_{es}$.

## 4 Experiments

### 4.1 Experimental Settings

The goal of the experiments is to compare DANCE with the baselines across all sub-cases of Universal DA (i.e., CDA, PDA, ODA, and OPDA) under the four object classification datasets and four settings for each dataset. We follow the settings of CDAN [25] for closed (CDA), SAN [2] for partial (PDA), STA [23] for open-set (ODA), and UAN [43] for open-partial domain adaptation (OPDA) in our experiments.

Table 2: Results on closed-set domain adaptation including SAFN [42], CDAN [25] and MDD [45].

| | Office (31 / 0 / 0) | | | | | | | Office-Home (65 / 0 / 0) | | | | | | | | | | | | | VisDA |
|---|---|---|---|---|---|---|---|---|---|---|---|---|---|---|---|---|---|---|---|---|---|---|
| **Universal comparison** | | | | | | | | | | | | | | | | | | | | | | |
| Method | A2W | D2W | W2D | A2D | D2A | W2A | Avg | A2C | A2P | A2R | C2A | C2P | C2R | P2A | P2C | P2R | R2A | R2C | R2P | Avg | 12/0/0 |
| SO | 74.1 | 95.3 | 99.0 | 80.1 | 54.0 | 56.3 | 76.5 | 37.0 | 62.2 | 70.7 | 46.6 | 55.1 | 60.3 | 46.1 | 32.0 | 68.7 | 61.8 | 39.2 | 75.4 | 54.6 | 46.3 |
| DANN [12] | 86.7 | 97.2 | 99.8 | 86.1 | **72.5** | **72.8** | **85.9** | 46.8 | 68.4 | 76.6 | 54.7 | 63.9 | 69.7 | 57.1 | 44.7 | 75.7 | 64.9 | 51.3 | 78.7 | 62.7 | 69.1 |
| ETN [3] | 87.9 | **99.2** | **100** | 88.4 | 68.7 | 66.8 | 85.2 | 46.7 | 69.5 | 74.8 | 62.1 | 66.9 | 71.9 | 56.7 | 44.1 | 77.0 | 70.6 | 50.4 | 77.9 | 64.0 | 64.1 |
| STA [23] | 77.1 | 90.7 | 98.1 | 75.5 | 51.4 | 48.9 | 73.6 | 30.4 | 46.8 | 55.9 | 33.6 | 46.2 | 51.1 | 35.0 | 28.3 | 58.2 | 51.3 | 33.1 | 66.5 | 44.7 | 48.1 |
| UAN [43] | 86.5 | 97.0 | **100** | 84.5 | 69.6 | 68.7 | 84.4 | 45.0 | 63.6 | 71.2 | 51.4 | 58.2 | 63.2 | 52.6 | 40.9 | 71.0 | 63.3 | 48.2 | 75.4 | 58.7 | 66.4 |
| DANCE | **88.6** | 97.5 | **100** | **89.4** | 69.5 | 68.2 | 85.5 | **54.3** | **75.9** | **78.4** | **64.8** | **72.1** | **73.4** | **63.2** | **53.0** | **79.4** | **73.0** | **58.2** | **82.9** | **69.1** | **70.2** |
| **Methods tailored for Closed Domain Adaptation** | | | | | | | | | | | | | | | | | | | | | | |
| SAFN [42] | 88.8 | 98.4 | 99.8 | 87.7 | 69.8 | 69.7 | 85.7 | 52.0 | 71.7 | 76.3 | 64.2 | 69.9 | 71.9 | 63.7 | 51.4 | 77.1 | 70.9 | 57.1 | 81.5 | 67.3 | NA |
| CDAN [25] | 93.1 | 98.2 | 100 | 89.8 | 70.1 | 68.0 | 86.6 | 49.0 | 69.3 | 74.5 | 54.4 | 66 | 68.4 | 55.6 | 48.3 | 75.9 | 68.4 | 55.4 | 80.5 | 63.8 | 70.0 |
| MDD [45] | 94.5 | 98.4 | 100 | 93.5 | 74.6 | 72.2 | 88.9 | 54.9 | 73.7 | 77.8 | 60 | 71.4 | 71.8 | 61.2 | 53.6 | 78.1 | 72.5 | 60.2 | 82.3 | 68.1 | 74.6 |

Table 3: Results on partial domain adaptation including SAN [2] and IAFN [42].

| | Office-Caltech(10 / 21 / 0) | | | | | | | Office-Home(25 / 40 / 0) | | | | | | | | | | | | | VisDA |
|---|---|---|---|---|---|---|---|---|---|---|---|---|---|---|---|---|---|---|---|---|---|---|
| **Universal comparison** | | | | | | | | | | | | | | | | | | | | | | |
| Method | A2C | W2C | D2C | D2A | W2A | Avg | | A2C | A2P | A2R | C2A | C2P | C2R | P2A | P2C | P2R | R2A | R2C | R2P | Avg | (6 / 6 / 0) |
| SO | 75.4 | 70.7 | 68.5 | 80.4 | 84.6 | 75.9 | | 37.1 | 64.5 | 77.1 | 52.0 | 51.3 | 62.4 | 52.0 | 31.3 | 71.6 | 66.6 | 42.6 | 75.1 | 57.0 | 46.3 |
| DANN [12] | 41.9 | 42.7 | 43.4 | 41.5 | 41.5 | 42.2 | | 35.5 | 48.2 | 51.6 | 35.2 | 35.4 | 41.4 | 34.8 | 31.7 | 46.2 | 47.5 | 34.7 | 49.0 | 40.9 | 38.7 |
| ETN [3] | **88.9** | **92.3** | 92.9 | 95.4 | 94.3 | **92.8** | | 52.1 | **74.5** | 83.1 | 69.8 | 65.2 | 76.5 | 69.1 | 50.6 | 82.5 | 76.3 | 53.8 | 79.1 | 69.4 | 59.8 |
| STA [23] | 75.7 | 72.4 | 62.8 | 70.5 | 67.7 | 69.8 | | 35.0 | 55.2 | 59.7 | 37.5 | 48.4 | 53.5 | 36.0 | 32.2 | 59.9 | 54.3 | 38.5 | 64.6 | 47.9 | 48.2 |
| UAN [43] | 47.1 | 49.7 | 50.6 | 55.5 | 61.6 | 52.9 | | 24.5 | 35.0 | 41.5 | 34.7 | 32.3 | 32.7 | 32.7 | 21.1 | 43.0 | 39.7 | 26.6 | 46.0 | 34.2 | 39.7 |
| DANCE | 88.8 | 79.2 | 79.4 | 83.7 | 92.6 | 84.8 | | **53.6** | 73.2 | **84.9** | 70.8 | 67.3 | 82.6 | 70.0 | 50.9 | 84.8 | 77.0 | 55.9 | 81.8 | 71.1 | **73.7** |
| **Methods tailored for Partial Domain Adaptation** | | | | | | | | | | | | | | | | | | | | | | |
| SAN [2] | NA | NA | NA | 87.2 | 91.9 | NA | | 44.4 | 68.7 | 74.6 | 67.5 | 65.0 | 77.8 | 59.8 | 44.7 | 80.1 | 72.2 | 50.2 | 78.7 | 65.3 | NA |
| ETN [3] | 89.5 | 92.6 | 93.5 | 95.9 | 92.3 | 92.7 | | 59.2 | 77.0 | 79.5 | 62.9 | 65.7 | 75.0 | 68.3 | 55.4 | 84.4 | 75.7 | 57.7 | 84.5 | 70.5 | 66.0 |
| IAFN [42] | NA | NA | NA | NA | NA | NA | | 58.9 | 76.3 | 81.4 | 70.4 | 73.0 | 77.8 | 72.4 | 55.3 | 80.4 | 75.8 | 60.4 | 79.9 | 71.8 | 67.7 |

**Datasets**. As the most prevalent benchmark dataset, we use **Office** [32], which has three domains (Amazon (A), DSLR (D), Webcam (W)) and 31 classes. The second benchmark dataset **OfficeHome (OH)** [40] contains four domains and 65 classes. The third dataset **VisDA (VD)** [30] contains 12 classes from two domains: synthetic and real images. We provide an analysis of varying the number of classes using Caltech [14] and ImageNet [8] because these datasets contain a large number of classes. Let $L_s$ denotes a set of classes present in the source, $L_t$ denotes a set of classes present in the target. The class split in each setting ($|L_s \cap L_t|/|L_s - L_t|/|L_t - L_s|$) is shown in each table. We follow the experimental settings of [25, 2, 23, 43] for each split (see suppl. material for details).

**Evaluation**. We use the same evaluation metrics as previous works. In CDA and PDA, we simply calculate the accuracy over all target samples. In ODA and OPDA, we average the accuracy over classes including "unknown". For example, an average over 11 classes is reported in the Office ODA setting. We run each experiment three times and report the average result.

**Implementation**. All experiments are implemented in Pytorch [29]. We employ ResNet50 [17] pre-trained on ImageNet [8] as the feature extractor in all experiments. We remove the last linear layer of the network and add a new weight matrix to construct **W**. For baselines, we use their implementation. Hyper-parameters for each method are tuned on the "Amazon to DSLR" OPDA setting. We set $\lambda$ in Eq. 9 as 0.05 and $m$ in Eq. 7 as 0.5 for our method. For all comparisons, we use the same hyper-parameters, batch-size, learning rate, and checkpoint. The analysis of sensitivity to hyper-parameters is discussed in the supplementary.

**Comparisons**. We show two kinds of comparisons to provide better empirical insights. The first is the universal comparison to the 5 baselines including state-of-the-art methods on CDA, PDA, ODA, and OPDA. As we assume that we do not have prior knowledge of the category shift in the target domain, so all methods use fixed hyper-parameters, which are tuned on the "Amazon to DSLR" OPDA setting. This means that even for CDA and PDA, all methods use "unknown" example rejection during testing. Since methods for CDA and PDA are not designed with "unknown" example rejection, we reject samples using the entropy of classifier's output. The second is the comparison with methods tailored for each setting. In addition to the 5 baselines, we report published state-of-the-art results on each setting if available. Note that the universal results should not be directly compared with the methods tailored for each setting, as they are optimized for each setting with prior knowledge and do not have "unknown" example rejection in CDA and PDA.

**Universal comparison baselines: Source-only (SO).** The model is trained with source examples without using target samples. By comparing to it, we can see how much gain we can obtain by performing adaptation. **Closed-set DA (CDA)**. Since this is the most popular setting of domain adaptation, we employ DANN [11], a standard approach of feature distribution matching between

Table 4: Results on open-set domain adaptation.

| Method | Office(10 / 0 / 11) | | | | | | | Office-Home(15 / 0 / 50) | | | | | | | | | | | | VisDA |
| | A2W | D2W | W2D | A2D | D2A | W2A | Avg | A2C | A2P | A2R | C2A | C2P | C2R | P2A | P2C | P2R | R2A | R2C | R2P | Avg | (6 / 0 / 6) |
|---|---|---|---|---|---|---|---|---|---|---|---|---|---|---|---|---|---|---|---|---|---|
| SO | 83.8 | 95.3 | 95.3 | 89.6 | 85.6 | 84.9 | 89.1 | 55.1 | 79.8 | 87.2 | 61.8 | 66.2 | 76.6 | 63.9 | 48.5 | 82.4 | 75.5 | 53.7 | 84.2 | 69.6 | 43.3 |
| DANN [12] | 87.6 | 90.5 | 91.2 | 88.7 | 87.4 | 87.0 | 88.7 | 62.1 | 78.0 | 86.4 | 75.5 | 72.0 | 79.3 | 68.8 | 52.5 | 82.7 | 76.1 | 58.0 | 82.7 | 72.8 | 48.2 |
| ETN [3] | 86.7 | 90.0 | 90.1 | 89.1 | 86.7 | 86.6 | 88.2 | 58.2 | 79.9 | 85.5 | 67.7 | 70.9 | 79.6 | 66.2 | 54.8 | 81.2 | 76.8 | 60.7 | 81.7 | 71.9 | 51.7 |
| STA [23] | 91.7 | 94.4 | 94.8 | 90.9 | 87.3 | 80.6 | 89.9 | 56.6 | 74.7 | 86.5 | 65.7 | 69.7 | 77.3 | 63.4 | 47.8 | 81.0 | 73.6 | 57.1 | 78.8 | 69.3 | 51.7 |
| UAN [43] | 88.0 | 95.8 | 94.8 | 88.1 | 89.9 | 89.4 | 91.0 | 63.3 | 82.4 | 86.3 | 75.3 | 76.2 | 82.0 | 69.4 | 58.2 | 83.4 | 76.1 | 60.5 | 81.9 | 74.6 | 50.0 |
| DANCE | 93.6 | 97.0 | 97.1 | 95.7 | 91.0 | 90.3 | 94.1 | 64.1 | 84.1 | 88.3 | 76.7 | 80.7 | 84.9 | 77.6 | 62.7 | 85.4 | 80.8 | 65.1 | 87.1 | 78.1 | 65.3 |

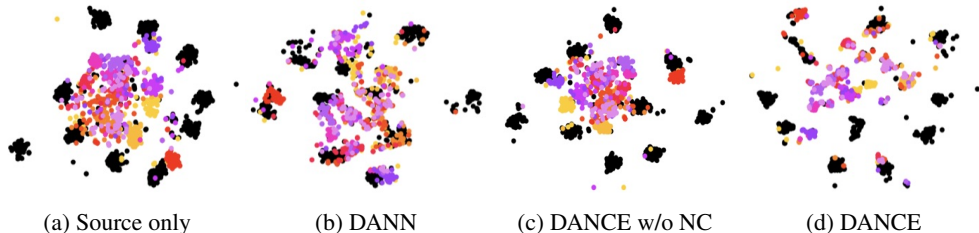

(a) Source only     (b) DANN     (c) DANCE w/o NC     (d) DANCE

Figure 3: t-SNE [26] plots of target examples (Best viewed in color). Black dots are target "known" examples while other colors are "unknown" examples. The colors indicate different classes. DANCE extracts discriminative features for "known" examples while keeping "unknown" examples far from "known" examples. Although we do not give supervision on "unknown" classes, some of them are clustered correctly.

domains. **Partial DA (PDA)**. ENT [3] is the state-of-the-art method in PDA. This method utilizes the importance weighting on source samples with adversarial learning. **Open-set DA (ODA)**. STA [23] tries to align target "known" examples as well as rejecting "unknown" samples. This method assumes that there is a particular number of "unknown" samples and rejects them as "unknown". **Open-Partial DA (OPDA)**. UAN [43] tries to incorporate the value of entropy to reject "unknown" examples.

## 4.2 Results

**Overview (Table 1)**. As seen in Table 1, which summarizes the universal comparison, DANCE is the only method which improves the performance compared to SO, a model trained without adaptation, in all settings. In addition, DANCE performs the best on open set and open-partial adaptation in all settings and the partial and closed domain adaptation setting for OfficeHome and VisDA. Our average performance is much better than other baselines with respect to both accuracy and rank. Due to a limited space, we put the result of OPDA in supplemental material. **CDA (Table 2)**. DANCE significantly improves performance compared to the source-only model (SO), and shows comparable performance to some of the baseline methods. Even compared to methods specialized in this setting, DANCE shows superior performance in OfficeHome. Some baselines show better performance in Office. However, such methods designed for CDA fail in adaptation when there are "unknown" examples. **PDA (Table 3)**. DANCE significantly improves accuracy compared to SO and achieves a comparable performance to ETN, which is the one of the state-of-the-art methods in PDA. Although ETN in the universal comparison shows better performance than DANCE in Office, it does not perform well on ODA and OPDA. In the case of VisDA and OfficeHome, DANCE outperforms all baselines. **ODA (Table 4)**. DANCE outperforms all the other baselines including ones tailored for ODA. STA and UAN are designed for the ODA and OPDA achieve decent performance on these settings but show poor performance on some settings in CDA and PDA. One reason is that their method assumes that there there is a particular number of "unknown" examples in the target domain and reject them as "unknown".

**Feature Visualization**. Fig. 3 shows the target feature visualization with t-SNE [26]. We use the ODA setting of "DSLR to Amazon" on Office. The target "known" features (black dots) are well clustered with DANCE. In addition, most of the "unknown" features (the other colors) are kept far from "known" features and "unknown" examples in the same class are clustered together. Although we do not give supervision on the "unknown" classes, similar examples are clustered together. The visualization supports the results of the clustering performance (see below).

**Ablation by clustering "unknown" examples**. Here, we evaluate how well the learned features can cluster samples of both "known" and "unknown" classes in the ODA setting. The goal of the ODA is to classify samples from "unknown" classes into a single class, i.e. "unknown". But, the "unknown" classes actually consist of multiple classes. Here, we evaluate the ability to cluster the "unknown" classes into their original classes. We train a new linear classifier on top of the fixed learned feature

Table 5: Linear classification accuracy given 1 labeled target sample per class in open-set setting (Known Accuracy/ Novel Accuracy). NC provides better clustered features for both known and novel classes. Adding entropy separation loss (DANCE) further improves the performance.

| Method | R2A known / novel | R2C known / novel | R2P known / novel | P2A known / novel | P2C known / novel | P2R known / novel |
|---|---|---|---|---|---|---|
| ImgNet | 37.5 / 31.0 | 35.3 / 36.4 | 64.8 / 56.9 | 36.9 / 31.0 | 36.3 / 36.0 | 66.3 / 45.5 |
| SO | 42.4 / 30.7 | 43.4 / 33.8 | 69.9 / 53.8 | 38.6 / 30.1 | 37.0 / 32.2 | 65.1 / 39.1 |
| DANN | 41.3 / 30.2 | 42.4 / 33.4 | 62.8 / 50.7 | 41.6 / 28.9 | 40.1 / 31.6 | 67.2 / 38.8 |
| NC | 48.4 / **33.9** | 47.8 / **36.6** | **74.9** / 56.6 | 45.6 / 33.3 | 42.5 / 37.5 | **74.7 / 45.6** |
| DANCE | **49.1** / 33.8 | **48.7** / 36.5 | **74.9 / 57.9** | **46.4 / 35.2** | **43.0 / 38.1** | 74.1 / 45.2 |

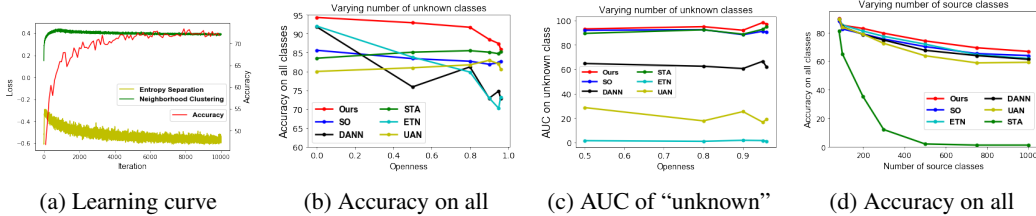

(a) Learning curve     (b) Accuracy on all     (c) AUC of "unknown"     (d) Accuracy on all

Figure 4: (a): Learning curve in the VisDA PDA setting. The accuracy improves as the two losses decrease. (b)(c): Increasing number of "unknown" classes. (d): Increasing number of source classes. DANCE outperforms than other methods even when there are many outlier source classes or many outlier target classes.

extractor. We use one labeled example per category for training. Since the feature extractor is fixed, we can evaluate its own ability to cluster the samples using the classifier. In this experiment, we use OfficeHome (15 "known" and 50 "unknown" classes). As we can see in Table 5, DANCE maintains or improves performance for "unknown" classes compared to the ImageNet trained model while the other baseline methods much worsen performance to classify "unknown" classes. NC provides well-clustered features for both types of classes and adding entropy separation further improves the performance. The result also shows the effectiveness of our method for class-incremental domain adaptation [21]. This result and the feature visualization indicate that the features learned by DANCE cluster samples from "unknown" classes well.

**The number of "unknown" classes.** We analyze the behavior of DANCE under the different the number of "unknown" classes. In this analysis, we use open set adaptation from Amazon in Office to Caltech, where there are 10 shared classes and many unshared classes. Openness is defined as $1 - \frac{|L_s \cap L_t|}{|L_t - L_s|}$. $L_s \cap L_t$ corresponds to the shared 10 categories. We increased the number of "unknown" categories, i.e. $|L_t - L_s|$. Fig. 4b shows the accuracy of all classes whereas Fig. 4c shows area under the receiver operating characteristic curve on "unknown" classes. As we add more "unknown" classes, the performance of all methods decreases. However, DANCE consistently performs better than other methods and is robust to the number of "unknown" classes.

**The number of source private classes.** We analyze the behavior under the different the number of source private classes in the OPDA setting. We vary the number of classes present only in the source (i.e., $|L_s - L_t|$). To conduct an extensive analysis, we use ImageNet-1K [8] as the source domain and Caltech-256 as the target domain. They have 84 shared classes. We use all of the unshared classes of Caltech as a "unknown" target while we increase the number of the classes of ImageNet (i.e., $|L_s - L_t|$). The result is shown in Fig. 4d. As we have more unshared source classes, the performance degrades as seen in Fig. 4d. DANCE consistently shows better performance. Since STA just tries to classify almost all target examples as "unknown," the performance is significantly worse.

## 5 Conclusion

In this paper, we introduce Domain Adaptive Neighborhood Clustering via Entropy optimization (DANCE) which performs well on universal domain adaptation. We propose two novel self-supervision based components: neighborhood clustering and entropy separation which can handle arbitrary category shift. DANCE is the only model which outperforms the source-only model in all settings and the state-of-the-art baselines in many settings. In addition, we show that DANCE extracts discriminative feature representations for "unknown" class examples without any supervision on the target domain.

## Broader Impact

Our work is applicable to training deep neural networks with less supervision via knowledge transfer from auxiliary datasets. Modern deep networks outperform humans on many datasets given a lot of annotated data, such as in ImageNet. Our proposed method can help reduce the burden of collecting large-scale supervised data in many applications where large related datasets are available. The positive impact of our work is to reduce the data gathering effort for data-expensive applications. This can make the technology more accessible for institutions and individuals that do not have rich resources. It can also help applications where data is protected by privacy laws and is therefore difficult to gather, or in sim2real applications where simulated data is easy to create but real data is difficult to collect. The negative impacts could be to make these systems more accessible to companies, governments or individuals that attempt to use them for criminal activities such as fraud. Furthermore, As with all current deep learning systems, ours is susceptible to adversarial attacks and lack of interpretability. Finally, while we show improved performance relative to state-of-the-art, negative transfer could still occur, therefore our approach should not be used in mission-critical applications or to make important decisions without human oversight.

## 6   Acknowledgement

This work was supported by Honda, DARPA and NSF Award No. 1535797.

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
