[Supplementary Material]

# Supplementary Material

## A  Dataset Detail

In PDA, 10 classes in Caltech-256 are used as shared classes $(L_s \cap L_t)$. The other 21 classes are used as source private classes $(L_s - L_t)$. Since DSLR and Webcam do not have many examples, we conduct experiments on D to A, W to A, A to C (Caltech), D to C, and W to C shifts. In OSDA, the same 10 classes are used as shared classes $(L_s \cap L_t)$ and the selected 11 classes are used as unknown classes $(L_t - L_s)$. The setting is the same as (11). In OPDA, the same 10 class are used as shared classes $(L_s \cap L_t)$ and then, in alphabetical order, the next 10 classes are used as source private classes $(L_s - L_t)$, and the remaining 11 classes are used as unknown classes $(L_t - L_s)$. The second benchmark dataset is **OfficeHome (OH)** (12), which contains four domains and 65 classes. In PDA, in alphabetical order, the first 25 classes are selected as shared classes $(L_s \cap L_t)$ and the rest classes are source private classes $(L_s - L_t)$. In OSDA, the first 15 classes are used as shared classes $(L_s \cap L_t)$ and the rest classes are used as unknown classes $(L_t - L_s)$. In OPDA, the first 10 classes are used as shared classes $(L_s \cap L_t)$, the next 5 classes are source private classes $(L_s - L_t)$ and the rest are unknown classes $(L_t - L_s)$. The third dataset is **VisDA** (9), which contains 12 classes from the two domains, synthetic and real images. The synthetic domain consists of 152,397 synthetic 2D renderings of 3D objects and the real domain consists of 55,388 real images. In PDA, the first 6 classes are used as shared classes $(L_s \cap L_t)$ and the rest are source private classes $(L_s - L_t)$. In OSDA, we follow (11) and use the 6 classes as shared classes $|L_s \cap L_t|$ and the rest as unknown classes $(L_t - L_t)$. In OPDA, the first 6 classes are shared classes $(L_s \cap L_t)$, the next 3 are source private classes $(L_s - L_t)$ and the other 3 classes are unknown classes $(L_t - L_s)$. We mainly perform experiments on these three datasets with four settings because it enables direct comparison with many state-of-the-art results. We provide an analysis of varying the number of classes using Caltech (5) and ImageNet (3) because these datasets contain a large number of classes.

## B  Implementation Detail

We list the implementation details which are excluded from the main paper due to a limit of space. We used TITAN X (Pascal) with 12GB. One GPU is used for each experiment and each experiment takes about 2 hours.

**DANCE (universal comparison).** The batch-size is set as 36. The temperature parameter in Eq. 5 is set as 0.05 by following (10). We train a model for 10,000 iterations with nestrov momentum SGD and report the performance at the end of the iterations. The initial learning rate is set as $0.01$, which is decayed with the factor of $(1 + \gamma \frac{i}{10,000})^{-p}$, where $i$ denotes the number of iterations and we set $\gamma = 10$ and $p = 0.75$. The learning rate of pre-trained layers is multiplied by $0.1$. We follow (10) for this scheduling method.

**Baselines (universal comparison).** We use the following released codes for ETN (1)(https://github.com/thuml/ETN), UAN (13)(https://github.com/thuml/Universal-Domain-Adaptation), and STA (7)(https://github.com/thuml/Separate_to_Adapt). We tune the hyper-parameter of these methods by validating the performance on OPDA, Amazon to DSLR, Office. Since we could not see improvements by changing the hyper-parameters from their codes, we employed the hyper-parameters provided in their codes. For ETN, we use the hyper-parameters for Office-Home. For UAN and STA, we use the hyper-parameters for Office. We implement DANN by ourselves and tuned the hyper-parameters by the performance on OPDA, Amazon to DSLR, Office. For all of these methods, we report the performance at the end of training for comparison. We observe that there is a gap in the performance between the best checkpoint and

Table A: Results on open-partial domain adaptation. USFDA (6) focuses on open-partial domain adaptation without access to source samples in adapting a model to a target domain. The number of UAN (13) in a lower row is taken from their paper.

| Method | Office(10 / 10 / 11) | | | | | | | Office-Home(10 / 5 / 50) | | | | | | | | | | | | | VisDA |
|---|---|---|---|---|---|---|---|---|---|---|---|---|---|---|---|---|---|---|---|---|---|
| | A2W | D2W | W2D | A2D | D2A | W2A | Avg | A2C | A2P | A2R | C2A | C2P | C2R | P2A | P2C | P2R | R2A | R2C | R2P | Avg | (6 / 3 / 3) |
| SO | 75.7 | 95.4 | 95.2 | 83.4 | 84.1 | 84.8 | 86.4 | 50.4 | 79.4 | 90.8 | 64.9 | 66.1 | 79.9 | 71.6 | 48.5 | 87.6 | 77.8 | 52.1 | 82.8 | 71.0 | 38.8 |
| DANN (4) | 87.6 | 90.5 | 91.2 | 88.7 | 87.4 | 87.0 | 88.7 | 59.9 | 80.6 | 89.8 | 77.5 | 73.3 | 86.4 | 78.5 | 61.5 | 88.5 | 80.3 | 62.1 | 82.4 | 76.7 | 50.6 |
| ETN (1) | 89.1 | 90.6 | 90.9 | 86.3 | 86.4 | 86.5 | 88.3 | 58.2 | 78.5 | 89.1 | 77.2 | 69.3 | 87.5 | 77.0 | 56.0 | 88.2 | 77.5 | 58.4 | 83.0 | 75.0 | 66.6 |
| STA (7) | 85.2 | 96.3 | 95.1 | 88.1 | 87.9 | 86.0 | 89.8 | 54.8 | 76.6 | **91.2** | 71.5 | 71.8 | 82.0 | 70.7 | 50.1 | 88.2 | 74.1 | 60.0 | 80.5 | 72.6 | 47.4 |
| UAN (13) | 76.2 | 82.0 | 80.4 | 80.0 | 93.8 | 92.2 | 84.1 | 60.8 | 79.1 | 87.8 | 72.4 | 73.5 | 83.2 | 78.6 | 56.4 | 87.4 | 79.9 | 61.1 | 79.8 | 75.0 | 47.3 |
| **DANCE** | **92.8** | **97.8** | **97.7** | **91.6** | **92.2** | **91.4** | **93.9** | **64.1** | **84.3** | 91.2 | **84.3** | **78.3** | **89.4** | **83.4** | **63.6** | **91.4** | **83.3** | **63.9** | **86.9** | **80.4** | **69.2** |
| *Methods tailored for Open-Partial Domain Adaptation* | | | | | | | | | | | | | | | | | | | | | |
| UAN (13) | 85.6 | 94.8 | 98.0 | 86.5 | 85.5 | 85.1 | 89.2 | 63.0 | 82.8 | 87.9 | 76.9 | 78.7 | 85.4 | 78.2 | 58.6 | 86.8 | 83.4 | 63.2 | 79.4 | 77.0 | 60.8 |
| USFDA (6) | 85.6 | 95.2 | 97.8 | 88.5 | 87.5 | 86.6 | 90.2 | 63.4 | 83.3 | 89.4 | 71.0 | 72.3 | 86.1 | 78.5 | 60.2 | 87.4 | 81.6 | 63.2 | 88.2 | 77.0 | 63.9 |

Table B: Evaluation on two metrics on open-set and open partial domain adaptation. OS is average of all classes. OS* is the average of known classes.

| | Open Set | | | | | | | | | | | |
|---|---|---|---|---|---|---|---|---|---|---|---|---|
| Method | A to W | | D to W | | W to D | | A to D | | D to A | | W to A | |
| | OS | OS* | OS | OS* | OS | OS* | OS | OS* | OS | OS* | OS | OS* |
| SO | 83.8 | 87.7 | 95.3 | 99.0 | 95.3 | 100.0 | 89.6 | 93.6 | 85.6 | 86.3 | 84.9 | 88.2 |
| DANN | 87.6 | 95.7 | 90.5 | 99.3 | 91.2 | 100.0 | 88.7 | 96.9 | 87.4 | **95.4** | 87.0 | 95.2 |
| ETN | 86.7 | 95.4 | 90.0 | 99.0 | 90.1 | 99.1 | 89.1 | 98.0 | 86.7 | 95.3 | 86.6 | 95.3 |
| STA | 91.7 | 94.6 | 94.4 | 98.1 | 94.8 | 100.0 | 90.9 | 94.2 | 87.3 | 88.8 | 80.6 | 82.4 |
| UAN | 86.2 | 94.6 | 89.5 | 98.5 | 90.2 | 99.2 | 89.8 | 98.7 | 85.8 | 94.4 | 84.2 | 92.7 |
| DANCE | **93.6** | **97.2** | **97.0** | **100.0** | **97.1** | **100.0** | **95.7** | **98.4** | **91.0** | 94.9 | **90.3** | **95.6** |
| | Open Partial | | | | | | | | | | | |
| Method | A to W | | D to W | | W to D | | A to D | | D to A | | W to A | |
| | OS | OS* | OS | OS* | OS | OS* | OS | OS* | OS | OS* | OS | OS* |
| SO | 75.7 | 79.2 | 95.4 | 98.1 | 95.2 | 100.0 | 83.4 | 88.3 | 84.1 | 84.9 | 84.8 | 85.7 |
| DANN | 83.0 | 90.0 | 89.3 | 97.1 | 89.5 | 96.9 | 81.9 | 88.8 | 80.2 | 86.8 | 78.2 | 77.8 |
| ETN | 89.1 | **98.0** | 90.6 | **99.7** | 90.9 | **100.0** | 86.3 | **94.9** | 86.4 | **95.0** | 86.5 | **95.1** |
| STA | 85.2 | 87.8 | 96.3 | 99.1 | 95.1 | 100.0 | 88.1 | 90.6 | 87.9 | 88.7 | 86.0 | 87.1 |
| UAN | 78.8 | 86.7 | 83.5 | 91.9 | 84.9 | 93.4 | 77.5 | 85.3 | 75.7 | 83.3 | 75.6 | 83.1 |
| DANCE | **92.8** | 96.4 | **97.8** | 99.1 | **97.7** | 99.6 | **91.6** | 94.0 | **92.2** | 94.5 | **91.4** | 94.7 |

the final checkpoint. This can explain the gap between the reported performance in their paper and the performance in our universal comparisons.

**Baselines tailored for each category shift.** We run experiments for ETN (A2C, W2C, D2C, PDA) since the results are not available in their papers. For ETN, we report the performance which employs the same hyper-parameters as the universal comparison but does not use "unknown" sample rejection. For the methods tailored for each setting, we show the performance of the results reported in their papers. "NA" indicates the results are not available in their paper. We observe the performance gap in our universal comparison and the reported performance in each paper. For example, the performance of UAN in OPDA has a big gap between the universal comparison and the reported accuracy although we use the same hyper-parameters. We could obtain similar performance to the reported number if we pick up the best checkpoint for each setting. But, we report the performance of fixed iterations' checkpoints for a fair comparison, which can explain the gap.

Table C: Standard deviation of DANCE in experiments on Office and VisDA. The deviation is calculated by three runs. DANCE shows descent deviations.

| Setting | A2W | D2W | W2D | A2D | D2A | W2A | Avg | VisDA |
|---|---|---|---|---|---|---|---|---|
| CDA | 88.6±0.4 | 97.5±0.4 | 100±0.0 | 89.4±1.3 | 69.5±1.5 | 68.2±0.0 | 85.5±0.2 | 70.2±0.3 |
| ODA | 93.6±2.3 | 97.0±0.2 | 97.1±0.5 | 95.7±0.3 | 91.0±0.8 | 90.3±0.2 | 94.1±2.5 | 65.3±2.3 |
| OPDA | 92.8±0.2 | 97.8±0.6 | 97.7±0.5 | 91.6±1.9 | 92.2±0.1 | 91.4±0.4 | 93.9±0.3 | 69.2±0.6 |

| Setting | A2C | W2C | D2C | D2A | W2A | Avg | VisDA | |
|---|---|---|---|---|---|---|---|---|
| PDA | 88.8±0.4 | 79.2±0.3 | 79.4±0.3 | 83.7±3.3 | 92.6±0.5 | 84.8±1.5 | 73.7±2.9 | |

Table D: Comparison between jigsaw (2; 8) and DANCE on the Office dataset. For a fair comparison, we replace the loss of entropy similarity with jigsaw puzzle loss.

| Setting | Method | A2W | D2W | W2D | A2D | D2A | W2A | Avg |
|---------|--------|------|------|-------|------|------|------|------|
| Closed | Jigsaw | 87.7 | **98.7** | **100.0** | 84.5 | 61.7 | 62.5 | 82.5 |
| | DANCE | **88.6** | 97.5 | **100.0** | **89.4** | **69.5** | **68.2** | **85.5** |
| Open | Jigsaw | 89.4 | 95.5 | 93.6 | 93.8 | 90.3 | 89.3 | 92.0 |
| | DANCE | **93.6** | **97.0** | **97.1** | **95.7** | **91.0** | **90.3** | **94.1** |

(a) Value of $\lambda$ in Eq. 9.    (b) Value of $m$ in Eq. 7.    (c) Value of $\rho$ in Eq. 7.

Figure A: (a): Varying the value of $\lambda$ in Eq. 9. (b): Varying the value of margin in Eq. 7. (c): Varying the value of $\rho$ in Eq. 7, which is determined based on the number of known classes.

## C  Supplemental Results

**Detailed results of ODA and OPDA.** Table A and Table B shows the detailed results of ODA and OPDA. OS* shows the averaged accuracy over known classes while OS shows the averaged accuracy including unknown class. DANCE shows good performance on both metrics. ETN shows better results on OS* than DANCE in several scenarios. In ETN results, OS* shows much better results on OS, which means that ETN is not good at recognizing unknown samples as unknown. This is clearly shown in Fig. 4 (c) in our main paper.

**Comparison with Jigsaw (2).** Table D shows the comparison with jigsaw puzzle based self-supervised learning. To consider the self-supervised learning part of DANCE, we replaced neighborhood clustering loss with the jigsaw puzzle loss on the target domain. The jigsaw puzzle loss is calculated on target samples. We can see that DANCE performed better in almost all settings and confirm the effectiveness of clustering based self-supervision for this task.

**Results with standard deviations.** Table C show results of DANCE with standard deviations. We show only the averaged accuracy over three runs in the main paper due to a limit of space. We show the standard deviation. We can observe that DANCE shows decent standard deviations.

**Sensitivity to hyper-parameters.** In Fig. A, we show the sensitivity to hyper-parameters on OPDA setting of Amazon to DSLR, which we used to tune the hyper-parameters. Although $\rho$ in Eq. 5 is decided based on the number of source classes, we show the behavior of our method when changing it in Fig. A(c). When we increase the value, more examples will be decided as known, then the performance on unknown examples decreases.