[Reviews · NeurIPS 2020]

Review 1

Summary and Contributions: This paper proposed a model named DANCE(Domain Adaptive Neighborhood Clustering vid Entropy optimization) to deal with the problem of universal domain adaptation. The contributions can be listed as follows: (i) propose a domain adaptation framework that can be used for universal domain adaptation problem (ii) design two novel loss function, neighborhood clustering and entropy separation, for such shift-agnostic adaptation problem (iii) learn discriminative features of 'unknown' target samples without supervision.

Strengths: This paper designs neighborhood clustering and entropy separation loss to better deal with the universal domain adaptation problem, which is a novel attempt. Experimental results shows that indeed DANCE outperform other methods, especially methods focusing on CDA, ODA, PDA and OPDA.

Weaknesses: The methods proposed are mainly based on empirical observation. However, DA problem is a machine learning problem rather than a simple computer vision problem, some theoretical proof or explanation will be welcomed. I think it will be better to submit this paper to a conference focusing on application rather than NeurIPS.

Correctness: Yes

Clarity: Yes

Relation to Prior Work: Yes

Reproducibility: Yes

Additional Feedback:


Review 2

Summary and Contributions: This work proposed a method called Domain Adaptative Neighborhood Clustering via Entropy optimization (DANCE) for universal domain adaptation, which can handle arbitrary category shift. DANCE proposes a novel neighborhood clustering technique to move each target sample either to a known class prototype in the source domain or to its neighbor in the target domain. Besides, an entropy-based feature alignment and rejection are proposed to align target features with the source, or reject them as unknown categories based on their entropy.

Strengths: This work proposes a simple and effective idea for universal domain adaptation, which is practical. The idea of clustering each target sample either to a known class prototype in the source domain or to its neighbor in the target domain is reasonable. The experimental comparisons are extensive, including experiments on multiple datasets, comparison with different types of methods, ablation study etc. The average performance on close-set DA, partial DA and openset DA is much better than existing methods, with a large margin. The paper is written clearly.

Weaknesses: I have only one concern: For a target sample belonging to the known category, a prototype in source and a neighbor in target may be both close to the target sample or even the neighbor in target may be closer to the target than the prototype. How does this affect the performance?

Correctness: yes, all claims and method are clear and correct.

Clarity: this work is written well and clear.

Relation to Prior Work: clearly

Reproducibility: Yes

Additional Feedback: ============== Post-rebuttal comments ============== The authors clearly answer my main concern but not explain whether it affects the performance. Considering the idea of this work is interesting and effective, i'd like to keep my score.


Review 3

Summary and Contributions: In this paper, the authors propose a method for what they call "universal" domain adaptation (closed, open, partial and mix of open and partial DA). The method, called DANCE, include two entropy-based constraints on top of domain-specific BN: (i) neighbourhood clustering and (ii) entropy separation. The authors show extensive experimental results on multiple DA settings and datasets.

Strengths: + The idea of having a single model to deal with different DA scenarios (open, close, partial and open/partial) without any prior knowledge about the category distribution is interesting, challenging and not much explored yet. + The paper has an extensive empirical evaluation, with results in different DA settings and on different domain-shift scenarios. + The proposed model achieves good results in many of the tasks evaluated

Weaknesses: - The paper has basically two main novel contributions: NC and ES. I found that both losses are designed a bit ad-hoc and more intuition on why they are designed as they are and why they work are necessary. For instance, it is not entirely clear to my why Eq. 6 would converge to clusters where each target point to known class prototype OR to its neighbour in the target. The choice for ES loss also seems ad-hoc with the \rho and m hyperparameters and not much analyses of why that loss was chosen and why does it converge to a reasonable solution. - Although I appreciate the effort of authors to validate the proposed approach, I find the experimental section difficult to follow. It is way too dense and contain way too much datasets/DA scenarios. It becomes very difficult to really see why/how the method is better than others. Specially because in most cases, the comparison between DANCE and other approaches is not really apple-to-apples. The experimental section would maybe be more readable if the authors would consider less subset fo experiments and focus more on understanding why each component of the proposed approach is useful (there is only one small table providing this information) then blindly comparing with other methods.

Correctness: The paper seems to be be in general correct, although I believe more justification (and empirical evidence) showing why/how the method work would be very helpful (see Weaknesses).

Clarity: The writing of the paper could be improved for clarity, specially Section 3 and the figures. The experimental section is also difficult to follow given the large amount of information on the relatively small number of pages.

Relation to Prior Work: Yes.

Reproducibility: No

Additional Feedback: ============== Post-rebuttal comments ============== The authors answer my concerns and I am raising my score to 6. The proposed method seem to be useful in different settings, but I still think experimental section could be more clear. I also recommend the authors to include the comments/explanations provided by the rebuttal on the revised version of the manuscript.


Review 4

Summary and Contributions: This paper presents a concept of Domain Adaptative Neighborhood Clustering via Entropy optimization (DANCE) to achieve universal domain adaptation. Specifically, this paper proposes two novel self-supervision losses: neighborhood clustering and entropy separation. DANCE performs well under all settings and the whole framework is simple and clean. In addition, DANCE can extract discriminative feature representations for “unknown” class examples without any supervision on the target domain.

Strengths: 1. This paper focuses on a more general and uniform adaptation setting and proposes a uniform solution with two specifically designed losses. 2. Clear paper writings and clear algorithm illustration. 3. Sufficient experimental results support author’s claims. 4. Satisfying performance on multiple domain adaptation tasks.

Weaknesses: 1. Although authors integrate many components such as the memory bank updating and the confidence threshold, the main ideas of this paper, Neighborhood Clustering (NC) and entropy separation (ES), seem to be very straightforward. NC is not new [1]. [1] Li S, Chen D, Liu B, et al. Memory-based neighbourhood embedding for visual recognition[C]//Proceedings of the IEEE International Conference on Computer Vision. 2019: 6102-6111. 2. Will the memory bank F and V bring in lots of additional memory cost? If so, it would largely reduce the practicability of the proposed algorithm. Needs more explanation and discussions. 3. Missing reference ID in Table2-6.

Correctness: Yes.

Clarity: Yes.

Relation to Prior Work: Yes.

Reproducibility: Yes

Additional Feedback: It would be helpful to add some discussions on effects of the employing the Domain Specific Batch Normalization. Use a higher-resolution version of the left part figure in Figure 2. --------------------------------------------------------- Authors' feedback well addressed my concerns. I will raise my score to 7.

[Author Response · NeurIPS 2020]

Thanks for the helpful feedback. We will address the concerns.

**R1**. **Q1**. No theoretical proof and explanation. **A1**. Just the lack of proof or theoretical explanation should not be a
reason to reject a paper for NeurIPS. Domain adaptation papers without proofs have been accepted by NeurIPS (e.g.,
Domain Separation Networks, NeurIPS'16 and Transferable Normalization NeurIPS'19). Despite the existence of
theoretical papers for domain adaptation, they are not useful for the universal domain adaptation because they do not
provide good insight on how to deal with open categories.

**R2**. **Q1**. What if the known target sample is closer to other targets than to prototypes? **A1**. It depends on how close the
target sample is and how many other target samples are nearby. Since we have both ES and NC loss, the target sample
can be aligned to the source prototype even if the target is nearer to other targets than the prototype.

**R3**. **Q1**. Losses are ad-hoc. More intuition for each loss. **A1**. ES is a carefully designed pseudo-labeling loss
giving "known" or "unknown" label to target samples. We need to decide whether a sample is "known" or "unknown".
Importantly, we do not even know whether we have "unknown" samples in target domain for the universal domain
adaptation. Then, even though there are many "unknown" samples or none of them, the objective function for "unknown"
needs to work well on both scenarios. The entropy of a classifier output shows the confidence of the prediction. Large
entropy implies that the classifier is uncertain about the prediction for the sample and the value intuitively implies a
distance from source classes. Such distance should be effective metric for "unknown" score under different proportions
of "unknown" samples. We assume that target samples with entropy smaller than a threshold are all known samples
while other samples are unknown ones. The entropy of classifier output gets larger in the log scale when the source
domain has many more classes. Therefore, the threshold of the entropy ($\rho$) is set larger with the scale ($\frac{\log(C)}{2}$). We try
to select only confident samples using confidence value $m$. NC forces each target sample to be closer to its neighbors,
which results in discriminative features. For example, if the nearest neighbor of sample A is B while that of B is C, all A,
B, and C can be put closer. Of course, NC may not form very compact clusters as shown in Fig 3 (d), it does not require
to know the number of classes in the target and is suitable for universal domain adaptation. **Q2**. Effectiveness of each
module NC, ES. **A2**. Table A, Table 7, and Fig. 3(c) vs (d) show the ablation of NC and ES. Ablation study in Table A
(left) corresponds to Table 6 (paper), where we classify unknown samples into their original class given a fixed feature
extractor and one labeled target sample per class. To perform well, features have to be well-clustered. We provide
DANCE w/o ES and DANCE w/o NC results. The results show that NC extracts well-clustered features for unknown
classes. Only with ES (w/o NC), the accuracy is worse or comparable to source-only (SO). Fig. 3(c) vs (d) supports the
observation too. Left of Table A shows the ablation of open-set DA for VisDA. From this results, ES is effective for both
alignment of known samples and rejection of unknown samples. Combining ES and NC further boosted performance.
Table 7 (paper) shows that NC is not enough to ensure the performance since it does not consider the assignment of
each sample to source class. **Q3**. Comparison between DANCE and other approaches is not apples-to-apples. **A3**. The
main result is summarized in Table 1 (paper), where DANCE performs best in terms of averaged rank. As existing
baselines are tailored to specific category-shift, we first perform "universal comparison" where we do not have any prior
knowledge on the type of category-shift. This "universal comparison" between ours and SO, DANN, ETN, STA, UAN
is fair in that the hyper-parameters and checkpoints are validated in the same way (see "B. Implementation Detail" in the
supp. for details). We show that while the category shift settings we evaluate are all different and not directly comparable
(and have specially designed algorithms), our method consistently has the best performance (ranked first or close to
first), despite not being specifically tuned for each setting. This is a very powerful advantage in real-world settings.
40

**R4**. **Q1**. Straight forward method. NC is not
new. **A1**. We would like to correct the misunder-
standing. **NC is a new approach, which is to-**
**tally different from "Memory-based neighbor-**
**hood embedding for visual recognition". The**
**mentioned method is basically for supervised**
**learning or few-shot learning and does not**
**have a module to handle unlabeled samples.**
They attempted to aggregate the neighbourhood

| Method | D to A | W to A | R to P | R to C | VisDA | VisDA (OS/UNK) |
|--------|--------|--------|--------|--------|-------|----------------|
| SO | 53.5 | 54.1 | 58.2 | 20.7 | 72.3 | 43.3 / 28.5 |
| w/o NC | 52.0 | 51.6 | 59.5 | 20.3 | 73.1 | 54.7 / 36.0 |
| w/o ES | **57.5** | **58.1** | **64.5** | **23.1** | 75.1 | 60.2 / 52.2 |
| DANCE | **57.5** | 57.4 | 63.5 | 22.8 | **75.2** | **65.2 / 79.8** |

Table A: Left: Analysis of ability to cluster "unknown" class samples
given one labeled sample per class. Right: Ablation in open-set DA for
Visda. UNK shows the accuracy to reject unknown class samples.

information for the discriminative embedding. By contrast, NC does not have a module to aggregate the information. It
tries to make neighboring unlabeled target samples closer and calculates the loss in an unsupervised way. **Q2**. Cost
of memory? **A2**. Storing features in memory does not add much cost. All experiments were done with a single
12GB GPU. If we have many more samples, we can limit the number of samples to store. **Q3**. Missing reference in
Table 2-6. **A3**. We will add the reference in Table 2-6. **Q4**. Effect of domain-specific batch normalization. **A4**. The
technique we used is exactly the same as [3]. The effect is shown in Table 8 of their paper (maybe better to show some
results.). [1] "Memory-based neighborhood embedding for visual recognition". [2] "Unsupervised Feature Learning via
Non-Parametric Instance Discrimination". [3] "Semi-supervised domain adaptation via minimax entropy"

[Meta-Review · NeurIPS 2020]

This paper presents a domain adaptation framework that can handle arbitrary domain shift. The reviewers like the practicality of the proposed approach and the experiment results are convincing. There is a concern regarding the main theoretical basis of the proposed method (R1). I think that this is not a valid reason to reject a paper, although of course the paper would have been stronger if there was a theoretical basis for it. However, R3 also noted the lack of clear intuition regarding the proposed method. I strongly encourage the authors to at least provide more intuition. The authors have included some discussions around this in the author response. I expect them to be included and elaborated in the paper. I also encourage the authors to include other details provided in the author response based on reviewers' feedback to improve the final version of the paper.